# Design, Synthesis and Evaluation of Novel Derivatives of Curcuminoids with Cytotoxicity

**DOI:** 10.3390/ijms222212171

**Published:** 2021-11-10

**Authors:** Chen-Yin Chen, Jin-Cherng Lien, Chien-Yu Chen, Chin-Chuan Hung, Hui-Chang Lin

**Affiliations:** 1School of Pharmacy, China Medical University, No. 100, Sec. 1, Jingmao Road, Beitun Dist., Taichung 406040, Taiwan; cychen3009@gmail.com (C.-Y.C.); jclien@mail.cmu.edu.tw (J.-C.L.); qaz90006@gmail.com (C.-Y.C.); 2Department of Pharmacy, China Medical University Hospital, 2 Yude Road, Taichung 40447, Taiwan

**Keywords:** curcumin, curcuminoids, *Alnus*, diarylheptanoids, α,β-unsaturated β-diketone, anticancer activity, cell cycle, apoptosis, flow cytometry, gene expression, p53 pathway, GADD45B, molecular docking

## Abstract

Curcumin and curcuminoids have been discussed frequently due to their promising functional groups (such as scaffolds of α,β-unsaturated β-diketone, α,β-unsaturated ketone and β′-hydroxy-α,β-unsaturated ketone connected with aromatic rings on both sides) that play an important role in various bioactivities, including antioxidant, anti-inflammatory, anti-proliferation and anticancer activity. A series of novel curcuminoid derivatives (a total of 55 new compounds) and three reference compounds were synthesized with good yields using three-step organic synthesis. The anti-proliferative activities of curcumin derivatives were examined for six human cancer cell lines: HeLaS3, KBvin, MCF-7, HepG2, NCI-H460 and NCI-H460/MX20. Compared to the IC_50_ values of all the synthesized derivatives, most α,β-unsaturated ketones displayed potent anti-proliferative effects against all six human cancer cell lines, whereas β′-hydroxy-α,β-unsaturated ketones and α,β-unsaturated β-diketones presented moderate anti-proliferative effects. Two potent curcuminoid derivatives were found among all the novel derivatives and reference compounds: (*E*)-5-hydroxy-7-phenyl-1-(3,4,5-trimethoxyphenyl)hept-1-en-3-one (compound **3**) and (1*E*,4*E*)-1,7-bis(3,4,5-trimethoxyphenyl)hepta-1,4-dien-3-one (compound **MD12a**). These were selected for further analysis after the evaluation of their anti-proliferative effects against all human cancer cell lines. The results of apoptosis assays revealed that the number of dead cells was increased in early apoptosis and late apoptosis, while cell proliferation was also decreased after applying various concentrations of (*E*)-5-hydroxy-7-phenyl-1-(3,4,5-trimethoxyphenyl)hept-1-en-3-one (compound **3**) and (1*E*,4*E*)-1,7-bis(3,4,5-trimethoxyphenyl)hepta-1,4-dien-3-one (compound **MD12a**) to MCF-7 and HpeG2 cancer cells. Analysis of the gene expression arrays showed that three genes (GADD45B, SESN2 and BBC3) were correlated with the p53 pathway. From the quantitative PCR analysis, it was seen that (1*E*,4*E*)-1,7-bis(3,4,5-trimethoxyphenyl)hepta-1,4-dien-3-one (compound **MD12a**) effectively induced the up-regulated expression of GADD45B, leading to the suppression of MCF-7 cancer cell formation and cell death. Molecular docking analysis was used to predict and sketch the interactions of the GADD45B-α,β-unsaturated ketone complex for help in drug design.

## 1. Introduction

The natural product curcumin is a major component of turmeric (*Curcuma longa*), a spice plant in the ginger family (Zingiberaceae) growing in Asia and it contains several bioactive constituents with various medicinal properties, such as antioxidant and anti-inflammatory properties. These active compounds derived from turmeric are named curcuminoids, a type of diarylheptanoids that have a linear di-phenol structure and an α,β-unsaturated β-diketone moiety. The chemical structure of curcumin was first determined as diferuloylmethane in 1910 and synthesized in the laboratory in 1913. Curcumin was discovered as a mixture existing in the keto and enol forms in 1953. The analogs of curcumin include demethoxycurcumin (DMC), bisdemethoxycurcumin (BDMC), tetrahydrocurcumin (THC) and turmerone (Figure 1). In the past decade, many studies have reported that curcuminoid derivatives have anticancer activities, such as inhibition of cell proliferation, cell cycle arrest, apoptosis and cell death [1,2,3,4,5,6,7,8,9].

Another series of diarylheptanoid derivatives, such as yashabushiketol, isolated from the plant *Alnus firma* and persenones A and B from *Persea Americana*, with the specific functional fragment β′-hydroxy-α,β-unsaturated ketone moiety, have also been reported to show a high level of biological activity, demonstrating antioxidant activity, anti-inflammatory activity, cytotoxic activity, antibacterial activity and anticancer activity. The genus *Alnus* has been used as a folk medicine for the treatment of fever, inflammation, stomachache, diarrhea, hepatitis and rheumatic disease [10]. Additionally, the derivatives have been found to have antitumor and anticancer activity and could potentially be anticancer drugs with an important scaffold and pharmacophore [11,12,13,14,15,16,17]. Consequently, we aimed to carry out further research on the design and synthesis of novel β′-hydroxy-α,β-unsaturated ketones, α,β-unsaturated β-diketones and α,β-unsaturated ketones, as well as to evaluate the cytotoxicity of these new synthesized compounds against several human cancer cells.

## 2. Results and Discussion

### 2.1. Chemistry

Three aspects had to be considered in the design and synthesis of the curcuminoid derivatives: the synthesis of curcuminoid derivatives with various aromatic rings for ring A, the synthesis of the different scaffolds (β′-hydroxy-α,β-unsaturated ketone, α,β-unsaturated β-diketone and α,β-unsaturated ketone) of the curcuminoid derivatives and the synthesis of curcuminoid derivatives with different numbers of methoxy or hydroxy groups on each of the phenyl rings A and B, categorized as the **MD** series. Compounds **1**–**8**, **10**–**17 and MD1**–**12** were synthesized in two parts (Figure 1). The chemical structures of the synthesized compounds are shown in Table 1. For Part A, aldehydes (phenyl and the heterocyclic group furan, pyridine, pyrrole, thiophene and indole) dissolved in toluene were treated with Wittig reagent. The reaction was allowed to reflux for 2 h to give compounds **A1**–**16** (Appendix B). For Part B, the derivatives of phenyl-propanoic acid were prepared using a two-step reaction to give compounds **B1**–**4** (Appendix C). A solution of compounds **A1**–**16** in THF was treated with lithium diisopropylamide (LDA) at −78 °C under argon. Then, the **B1**–**4** –THF mixture was slowly transferred to the A1–15-THF solution at −78 °C under argon. The reaction was stirred for 1 h to give the corresponding β′-hydroxy-α,β-unsaturated ketones **1**–**8**, **10**–**17** and **MD1**–**12** in 60–90% yields (except for compound **8**, for which the yield was less than 60%) [18,19,20]. Compound **18** (with a chlorine substituent at the C5 beta position), a side product of compound **6,** might be obtained during the acid workup with ammonium chloride. The structures of the derivatives were determined by mass and by NMR analysis (Appendix A).

A solution of β′-hydroxy-α,β-unsaturated ketone in CH_2_Cl_2_ was cooled to 0 °C and treated with *p*-toluensulfonic acid (*p*-TSA) for 30 min to give the corresponding α,β-unsaturated ketones **1a**–**7a**, **12a**, **14a** and **MD1a**–**12a [21]**. The α,β-unsaturated ketones with a heterocyclic ring A (2-pyridine, 3-pyridine, 4-pyridine, 3-furan, 2-methyl-2-thiophene, 2-pyrrole and 3-indole) could not be synthesized in this reaction. For example, compound **9** was formed when the reaction was quenched by adding water. The double bonds were replaced by two hydroxy groups at the C1 and C5 beta positions.

Compounds **1**, **2, 3**, **4**, **6**, **7** and **MD4** were treated with the Dess–Martin reagent at 0 °C to room temperature for half an hour to give the corresponding α,β-unsaturated β-diketones **19**–**24** and **MD13** (Figure 2) [22]. Using NMR analysis, compounds **19**–**24** and **MD13** were detected in the enol form, which is interconvertible with the keto form. The enol form is more stable due to the presence of a continuous conjugated system and intramolecular hydrogen bonding [13] (Figure 3). Alterations of the chemical shifts in the ^1^H and ^13^C NMR spectra were observed. In the example of the new compound **20**, the proton 4 (H4) at the alpha’-carbon position was determined at 5.66 ppm in the ^1^H NMR spectra. The peak of the carbonyl group at C3 shifted from 201.9 ppm to the up-field at 179.0 ppm, whereas the peak of the beta’ carbon (C5) connected to the hydroxy group shifted from 68.9 ppm to the down-field at 201.1 ppm in the ^13^C NMR spectra (Appendix A).

### 2.2. Biological Data

#### 2.2.1. In Vitro Anti-Proliferative Activities

In total, 58 derivatives were synthesized, including compounds **1 [17]**, **19** and **MD7** as reference compounds. The anti-proliferative effects of the synthesized compounds **1**–**24**, **1a**–**7a**, **12a**, **14a**, **MD1**–**13** and **MD1a**–**12a** against six human cancer cells (HeLaS3, KBvin, MCF-7, HepG2, NCI-H460 and NCI-H460/MX20), are summarized in Table 2. In the summary of the IC_50_ values of the 58 compounds in Table 2, it can be seen that the IC_50_ values of the reference diarylheptanoid compound **1** against all six human cancer cell lines were over 40 μM. The IC_50_ values of derivatives **3**, **MD7a**, **MD8a** and **MD12a** among the 58 compounds against all six human cancer cell lines were dramatically lower, at about 1–9 μM. IC_50_ values below 10 μM among these 58 derivatives against more than three human cancer cells were found for compounds **2a**, **3**, **3a**, **MD2a**, **MD3a**, **MD4a**, **MD5a**, **MD6**, **MD6a**, **MD7**, **MD7a**, **MD8**, **MD8a**, **MD11**, **MD11a** and **MD12a**.

The IC_50_ data for curcuminoid derivatives showed that as the number of methoxy groups on the benzene ring (on both ring A and ring B) increased, the activity of the compounds tended to become greater, especially in the MD series. When the methoxy group on the benzene ring (ring A) of the diarylheptanoid derivatives was replaced with a hydroxyl group or a halogen group (such as fluorine or chlorine), the IC_50_ values of compounds **5a**, **6a** and **7a** against HelaS3, KBvin, MCF-7 and HepG2 were around 5–17 μM. When ring A, which is a benzene group, was replaced with different heterocyclic groups, the activity of the synthesized compounds (**8**–**17**, **12a** and **14a)** did not increase by much, except for that of compound **11** against HelaS3 and HepG2. Compared to reference compound **19**, the activity of synthesized compounds **21** and **22**, which have two and three methoxy groups on the benzene ring (ring A), only increased a little. The IC_50_ values of compounds 23 and 24, which have F and Cl, respectively, on the benzene ring (ring A), were all measured as over 40 μM. Furthermore, based on the IC_50_ values of α, β-unsaturated β-diketone (enol form), β′-hydroxy-α,β-unsaturated ketones and α, β-unsaturated ketones, the α,β-unsaturated ketones of curcuminoids are more active than the two other forms of the derivatives.

Comparing the IC_50_ values among the tested compounds (**1**–**24** and **1a**, **2a**, **3a**, **4a**, **5a**, **6a**, **7a**, **12a** and **14a**) against HeLaS3, the synthesized derivatives with a skeleton of α,β-unsaturated ketone were more potent than the derivatives with a skeleton of β′-hydroxy-α,β-unsaturated ketone or α,β-unsaturated β-diketone, except for compound **3**. Additionally, comparing the IC_50_ values of the ring A portion (where ring B was benzene) of the synthesized derivatives, it was found that ring A, when used as the phenyl group of the derivatives, was more active than when it was used as the heterocyclic aromatic group of the derivatives. In addition, when comparing the different substituents of the phenyl group of ring A (with ring B as benzene) of α,β-unsaturated ketones, it was found that compounds **1a**, **3a**, **5a**, **6a** and **7a** had similar levels of activity against HeLaS3. The IC_50_ values were measured to be about 5–6 μM. The halogen (F or Cl) and hydroxy groups on the ring A benzene (with ring B as benzene) of α,β-unsaturated ketones against HeLaS3 were shown to be a little sensitive. The IC_50_ values of these α,β-unsaturated ketones (**5a**, **6a** and **7a**) were measured as around 5.25–6.29 μM. Comparing the IC_50_ values for both ring A and ring B as benzenes of β′-hydroxy-α β-unsaturated ketones and α,β-unsaturated ketones against HeLaS3, it was found that most α,β-unsaturated ketones were more potent than β′-hydroxy-α,β-unsaturated ketones, except for compounds **MD2**, **MD5**, **MD6**, **MD7** and **MD11**, which were measured to have activities of 7, 6, 9.2, 5.6, 4.68 and 5.7 μM, respectively, whereas the activities of α,β-unsaturated ketones were similar (IC_50_ values were around 4–9 μM) except for those of compounds **MD1a** and **MD10a**, which were measured as 24.4 and 18.8 μM, respectively. As the number of methoxy groups on both ring A and ring B (as benzenes in the synthesized compounds) was increased, the activity displayed was slightly increased.

For the compounds tested against Kbvin, the IC_50_ values showed that the derivatives where ring A was benzene (where ring B was benzene) were more potent than those where ring A was a heterocyclic aromatic ring. Moreover, as the number of methoxy groups on ring A and ring B increased, their activities also increased, whereas the halogen (fluorine or chlorine) on ring A of the derivatives did not improve the activity against KBvin very much. The tested compounds that had a skeleton of α,β-unsaturated ketone were more active than the compounds with β′-hydroxy-α,β-unsaturated ketone and α,β-unsaturated β-diketone (enol form) against the Kbvin cancer cell line.

For the NCI-H460 cancer cell line, the IC_50_ values showed that compounds **3**, **MD7a**, **MD8a** and **MD12a** were more active than all the synthesized compounds against NCI-H460 cancer cells and these activities were measured as 7.25, 4.3, 5.2 and 4.5 μM, respectively. This means that derivatives that had ring A and ring B as benzene had high activity against the NCI-H460 cancer cell line. Additionally, the methoxy groups on both ring A and ring B of the derivatives were more potent than those among all the synthesized compounds against the NCI-H460 cancer cell line. Analyses of the activity showed that the derivatives with a skeleton of α,β-unsaturated ketone were more potent than the derivatives with a skeleton of β′-hydroxy-α,β-unsaturated ketone (except for compound **3**) or α,β-unsaturated β-diketone (enol form).

For the compounds tested against the NCI-H460MX20 cancer cell line, compounds **3**, **MD7**, **MD7a**, **MD8**, **MD8a**, **MD9a** and **MD12a** were found to be the most active of all the synthesized compounds. The IC_50_ values of the derivatives with a skeleton of β′-hydroxy-α,β-unsaturated ketone (compounds **3**, **13** and **15**) were 8.25, 2.5 and 9.7 μM, respectively. Compounds **14**, **16**, **18** and **24** with an α β-unsaturated ketone moiety were discovered to have remarkable activity and the IC_50_ values were recorded as 1.2, 2.7, 5.3 and 5.63 μM, respectively.

For the compounds tested against the MCF-7 cancer cell line, the IC_50_ values showed that the methoxy group on ring B (as benzene) of the derivatives played an important role in their activity. The data showed that most of the synthesized compounds of the **MD** series were potent against MCF-7 cancer cells, whereas only a few derivatives (compounds **2a**, **3**, **3a** and **5a**) that had different functional groups on ring A and ring B (as benzene only) were active against MCF-7 cancer cells. The IC_50_ values of compounds **2a**, **3**, **3a** and **5a** against MCF-7 cancer cells were determined to be 9.87, 7.59, 8.09 and 8.52 μM, respectively. The range of IC_50_ values of the **MD** series was measured to be about 4.6–8.8 μM and skeletons of α,β-unsaturated ketone and β′-hydroxy-α,β-unsaturated ketone in the **MD** derivatives had a similar activity against the MCF-7 cancer cells.

Comparing the IC_50_ values of the reference compounds (**1** and **19**) and the synthesized compounds against the HepG2 cancer cell line, it could be seen that compounds **2a**, **3** and **3a** (with ring B as benzene only), compound **11** (with ring A as pyridine), most of the **MD** series (**MD5a**, **MD6**, **MD6a**, **MD7**, **MD7a**, **MD8**, **MD8a**, **MD9a**, **MD11**, **MD11a** and **MD12a**) and **MD13** with a skeleton of α,β-unsaturated β-diketone (enol form) had better activities than the reference compounds. In the MD series, the activities of the derivatives with skeletons of α,β-unsaturated ketone and β′-hydroxy-α,β-unsaturated ketone against the HepG2 cancer cell line were similar. However, the methoxy group on ring A and ring B (with both ring A and ring B as benzenes) of the synthesized derivative was an important factor in enhancing the activity against HepG2 cancer cells.

Comparing the IC_50_ values of the synthesized derivatives with the skeletons of β′-hydroxy-α,β-unsaturated ketone, α,β-unsaturated ketone and α,β-unsaturated β-diketone (enol form), it was found that derivatives **3** and **MD12a** had considerable anti-proliferative activity against all six cancer cell lines. Compound **3** had an active β′-hydroxy-α,β-unsaturated ketone moiety connected with two phenyl groups: ring A (three methoxy groups) and ring B (benzene only). **MD12a** contained an α,β-unsaturated ketone moiety that was connected to a phenyl ring (three methoxy groups on each phenyl ring) on both sides.

#### 2.2.2. Cell Cycle and Apoptosis Analysis by Flow Cytometry

Based on the anti-proliferative activity data shown above, a cell cycle analysis was performed using an easyCyte 5 cytometer and the collected samples were examined with Annexin V/PI staining via flow cytometry. MCF-7 and HepG2 cells were treated with various concentrations (5, 10 and 20 μM) of compounds **3** and **MD12a** for 24 h and 48 h to determine the cell cycle phase (Figure 2a,b).

For MCF-7 cells treated with compound **3**, it was shown that compound **3** induced cell cycle arrest and a time-dependent increase in the G1 phase percentage. The percentages of untreated cells in the sub-G1, G1, S and G2/M phases were determined to be 8.67 ± 1.88, 48.1 ± 3.53, 17 ± 1.83 and 19.95 ± 6.85% as well as 5.72 ± 5.08, 63.73 ± 8.61, 12.8 ± 2.7 and 15.5 ± 1.37% for 24 h and 48 h, respectively. The percentages of MCF-7 cells treated with 5 μM of compound **3** in the sub-G1, G1, S and G2/M phases were determined to be 6.35 ± 0.48, 50.9 ± 0.98, 12.55 ± 2.6 and 19.1 ± 4.1% as well as 5.95 ± 4.91, 63.4 ± 10.0, 13.22 ± 5.04 and 14.6 ± 1.37% for 24 h and 48 h of treatment, respectively. The percentages of MCF-7 cells treated with 10 μM of compound **3** in the sub-G1, G1, S and G2/M phases were determined to be 7.95 ± 0.16, 50.1 ± 3.39, 12.9 ± 0.14 and 23.1 ± 10.88% as well as 6.35 ± 1.97, 60.1 ± 9.00, 13.1 ± 0.77 and 17.6 ± 8.95% for 24 h and 48 h of treatment, respectively. The percentages of MCF-7 cells treated with 20 μM of compound **3** in the sub-G1, G1, S and G2/M phases were evaluated to be 15.02 ± 10.57, 50.8 ± 7.07, 14.05 ± 5.02 and 14.12 ± 0.03% as well as 10.07 ± 5.69, 60.2 ± 8.92, 12.56 ± 2.66 and 14.96 ± 2.45% for 24 h and 48 h of treatment, respectively (Table 3).

For MCF-7 cells treated with **MD12a**, the detection analysis showed that **MD12a** induced a dose-dependent increase in the sub-G1 phase. Compared with untreated cells, MCF-7 cells treated with **MD12a** for 24 h were arrested at the sub-G1 phase and induced cell apoptosis. Even for 48 h of treatment, the percentage of treated cells in the sub-G1 phases was still increased. The accumulation of control cells in the sub-G1, G1, S and G2/M phases was determined as 8.67 ± 3.53, 48.1 ± 3.53, 17 ± 1.83 and 19.95 ± 6.85% as well as 5.72 ± 5.08, 63.73 ± 8.61, 12.84 ± 2.7 and 15.56 ± 1.37% for 24 h and 48 h, respectively. The percentage of MCF-7 treated with 5 μM of **MD12a** in the sub-G1, G1, S and G2/M phases was determined to be 14.35 ± 9.40, 53.2 ± 17.18, 8.8 ± 1.97 and 17 ± 3.81% as well as 28.06 ± 5.25, 44.93 ± 6.50, 18.03 ± 4.80 and 9.77 ± 6.05% for 24 h and 48 h treatment, respectively. The percentage of MCF-7 cells treated with 10 μM of **MD12a** in the sub-G1, G1, S and G2/M phases was calculated as 50.3 ± 25.88, 36.4 ± 17.88, 14.25 ± 3.18 and 4.55 ± 4.16% as well as 65 ± 8.20, 22.4 ± 9.00, 11.44 ± 2.75 and 6.45 ± 9.58% for 24 h and 48 h treatment, respectively. The percentage of MCF-7 treated with 20 μM of **MD12a** in the sub-G1, G1, S and G2/M phases was determined as 85.4 ± 4.38, 20.9 ± 4.17, 0.25 ± 0.016 and 0.30 ± 0.29% as well as 98 ± 1.83, 1.75 ± 1.78, 0.21 ± 0.36 and 1.30 ± 2.16% for 24 h and 48 h treatment, respectively. Moreover, the evidence shows that the distribution of MCF-7 cells treated with 10 μM and 20 μM of **MD12a** arrested at the sub-G1 peak of the cell cycle was much higher than for cells treated with 5 μM of **MD12a** (Table 3).

In Table 3, the results show that compound **3** slightly induced cell cycle arrest and apoptosis in HepG2 cells. The detection demonstrated a small time-dependent increase in the sub-G1 phase. The percentage of untreated HepG2 cells arrested at the sub-G1 phase was calculated as 1.39 ± 0.67 and 6.00 ± 3.86% for 24 h and 48 h, respectively, whereas the percentage of HepG2 treated with 5 μM, 10 μM and 20 μM accumulated in the sub-G1 phase was determined as 2.47 ± 1.78, 0.90 ± 0.55 and 0.96 ± 0.76% as well as 6.97 ± 6.86, 9.76 ± 11.72 and 8.08 ± 7.54% for 24 h and 48 h treatment, respectively.

In Table 3, **MD12a** induced HepG2 cell cycle arrest and apoptosis and the detection analysis represented a dose-dependent increase in the sub-G1 phase for 24 and 48 h of treatment. The profile also showed that **MD12a** induced a dose-dependent decrease in the G2/M phase. The percentage of untreated cells in the sub-G1 phase was evaluated as 1.39 ± 0.67% and 6.00 ± 3.86% for 24 h and 48 h, respectively, whereas the proportion of HepG2 cells treated with 5 μM, 10 μM and 20 μM arrested at the sub-G1 phase was calculated as 1.06 ± 0.56, 1.79 ± 1.41 and 3.90 ± 2.57% as well as 13.35 ± 14.34, 26.95 ± 7.8 and 75.95 ± 20.01% for 24 h and 48 h of treatment, respectively. The results showed that the treatment of HepG2 cells with 10 μM and 20 μM of **MD12a** led to a significant increase in the sub-G1 phase over 48 h of treatment.

For the cell flow cytometry analysis, where MCF-7 was treated with different concentrations (5 μM, 10 μM and 20 μM) of compounds **3** and **MD12a** for 24 h and 48 h, the images of the distribution of cell apoptosis in MCF-7 showed that the treatment of MCF-7 cells with compound **3** slightly increased the level of apoptosis, whereas the treatment of MCF-7 cells with **MD12a** induced a higher level of apoptosis as well as showing a significant dose-dependent increase in the level of apoptosis (Figure 3a,b).

When HepG2 cancer cells were treated with various concentrations (5 μM, 10 μM and 20 μM) of compounds **3** and **MD12a** for 24 h, flow cytometry images showed that compound **3** induced no significant change in cell apoptosis, whereas **MD12a** induced dose-dependent apoptosis in treated cells compared with untreated HepG2 cells (Figure 3c). When HepG2 cells were treated with different concentrations of compound **3** for 48 h, the images showed that compound **3** slightly increased the level of late apoptosis (Figure 3d). In contrast, **MD12a** induced a high level of apoptosis in HepG2 cells as the concentration of **MD12a** increased.

#### 2.2.3. Cell Morphology

In order to observe the drug response and activity of the cancer cells, their cell morphologies were recorded. Photographs of the morphological alterations of MCF-7 and HepG2 cancer cells treated with various concentrations (5 μM, 10 μM and 20 μM) of compound **3** and **MD12a** were taken using a ZOOMKOP ZK-2500 microscope under ×400 magnification (EC-H PL ×40) for 24 h and 48 h of treatment (Figure 4a–d). For both 24 h and 48 h, the images (Figure 4a,b) showed that MCF-7 cells treated with different concentrations of compound **3** and **MD12a** looked inflated, deformed, exfoliated and dead on the surface of the wells, whereas adherence of the control MCF-7 cells to the wells when not treated with the tested compounds was observed.

For HepG2 cells treated with compounds **3** and **MD12a** for 24 h and 48 h, the images (Figure 4c,d) displayed more deformed and exfoliated HepG2 cells when the concentration of both the tested compounds was increased, compared with the control group of HepG2 cells, as observed by a microscope under ×400 magnification.

#### 2.2.4. Gene Expression

##### Histogram Plot

The histogram plot displays the fold change distribution calculated using Rosetta Resolver 7.2 with the error model adjusted using Amersham Pairwise Ration software. Data are for all probes, excluding control and flagged probes (Figure 5).

##### Volcano Plot

The scatterplot displays the statistical significance on the y-axis (*p*-value) versus the magnitude of the change on the x-axis (fold change). The red dotted line shows the *p*-value cut-off (0.05), while the green dotted line shows the fold change cut-off (log2 |fold change| ≥ 1 and the negative logarithm of the *p*-value is shown as blue dots (<0.05)). Expression data were plotted for all probes, excluding the control and flagged probes. The plot shows that the most up-regulated genes are toward the right, whereas the most down-regulated genes are toward to the left (Figure 6).

##### Number of Differentially Expressed Genes

The number of differentially expressed genes (up- and down-regulated genes) for each comparison is shown in Table 4. There are four items for the comparison: 1. HepG2 cancer cells treated with compound **3** versus HepG2 control (cells treated with DMSO as control groups); 2. HepG2 cancer cells treated with compound **MD12a** versus HepG2 control; 3. MCF-7 cancer cells treated with compound **3** versus MCF-7 control; 4. MCF-7 cancer cells treated with compound **MD12a** versus MCF-7 control. Standard selection criteria to identify differentially expressed genes are as follows: (1) log2 |Fold change| ≥ 1 and *p* < 0.05 (2) log2 ratios = ”NA” and the differences of intensity between the two samples ≥ 1000.

##### Principal Component Analysis (PCA)

A PCA plot based on all genes in the microarray was constructed to evaluate the differences among the biological replicated and treatment conditions on both the MCF-7 and HepG2 cancer cells. The variable values of the first three principal components (PC1, PC2 and PC3) were evaluated as 72.48%, 17.81% and 5.70%, respectively (Figure 7).

##### Clustering Analysis

For an advanced data analysis based on the fold changes and *p*-values, the intensity data were evaluated to identify the genes expressed. The correlation of the expression profiles between the different samples and treatment conditions was performed using a clustering analysis. Up-regulated genes are represented in red and down-regulated genes are represented in green. The heatmap (Figure 8) shown below indicates the expression data for the gene list, which is labeled from left to right as follows: HG2C_H001, HG2C_H002, HG2A_H003, HG2A_H005, HG2B_H006, M7C_H007, M7C_H008, M7A_H009, M7AH010, M7B_H011 and M7B_H012 (note: HG2 = HepG2, M7 = MCF-7, C = control, A = compound **3** and B = **MD12a**).

#### 2.2.5. The p53 Signaling Pathway and Quantitative PCR (qPCR) Analysis for HepG2 and MCF-7 Cells Treated with Compounds **3** and **MD12a**

Based on the quantitative PCR data, Figure 9 displays the fold changes (x-axis) versus treatment groups (y-axis). The target genes were the following: GADD45B, SENSN2 and BBC3. The target genes were selected from the results of the gene expression analysis. The analysis of the mRNA expression showed that these three genes were up-regulated and associated with the p53 signaling pathway (Figure 10) for the treatment of MCF-7 and HepG2 cancer cells (Table 4). The reference gene used was GAPDH. The primer sequences are displayed in Table 5.

In Figure 9a, the fold changes versus the three target genes (GADD45B, SENSN2 and BBC3) for HepG2 cancer cells treated with compound **3** were evaluated as 1.69, 0.83 and 1.39, respectively. The fold changes indicated that compound **3** slightly induced GADD45B and BCC3 genes compared to the value of the control group, whereas the fold change of the SENSN2 gene was a little lower than that of the control. In Figure 9b, the fold changes versus the three target genes (GADD45B, SENSN2 and BBC3) for HepG2 cancer cells treated with **MD12a** were evaluated as 1.65, 0.83 and 1.24, respectively. The results also showed that the fold changes of the GADD45B and BBC3 genes were higher than the value of the control group.

In Figure 9c, the fold changes versus the three target genes (GADD45B, SENSN2 and BBC3) for MCF-7 cancer cells treated with compound **3** were evaluated as 0.83, 0.82 and 1.12, respectively. The data showed that only the fold change of the BBC3 gene was slightly increased, whereas the fold changes of GADD45B and SENSN2 were decreased compared with the value of the control group. In Figure 9d, the fold changes against the three target genes (GADD45B, SENSN2 and BBC3) for MCF-7 cancer cells treated with **MD12a** were evaluated as 5.04, 4.18 and 3.35, respectively. The results showed that the fold changes of these three target genes were higher than the values obtained for the control group. The application of the tested compound **MD12a** to both HepG2 and MCF-7 cancer cells was found to markedly induce the GADD45B gene with a high mRNA protein level compared to compound **3**. Thus, compound **MD12a** was found to sufficiently induce the GADD45B gene with up-regulated expression to suppress cancer cell formation and cause cell death in the p53 pathway.

#### 2.2.6. Docking Interaction of Compound **MD12a** with GADD45B

A molecular docking analysis was performed in order to provide computational predictions in two and three dimensions for the interactions between the proteins and the tested compounds (Figure 11a,b). The binding interaction of the ligand–**MD12a** complex was analyzed using Discovery Studio 2.5 software, which is used for docking calculations. For the docking simulation, the protein structure of GADD45B was downloaded from the AlphaFold Protein Structure Database [24,25]. The preparation was performed using the Prepare Protein module in Discovery Studio 2.5 to remove crystal water in the crystallographic structure, insert missing atoms into incomplete residues, protonate the structure of both proteins with the Chemistry at Harvard Macromolecular Mechanics (CHARMM) force field [26] and optimize the side-chain conformation for residues with inserted atoms. For GADD45B, four potential active sites in the protein cavities were identified with Discovery Studio 2.5 using a shape filter and Monte Carlo ligand conformation generation and optionally minimized with the CHARMM force field [27]. The interaction between **MD12a** and GADD45B included one hydrogen bonding interaction and three hydrophobic interactions. The oxygen on the carbonyl is the hydrogen bond acceptor that interacts with the residue of ARG115. The 3,4,5-methoxyphenyl moieties of **MD12a** are stabilized by the hydrophobic interaction within the drug binding site formed by the residues of ALA106, ALA114 and MET95.

## 3. Materials and Methods

All organic solvents and reagents were commercially available and purchased from Sigma-Aldrich (St. Louis, MO, USA), Acros Organics (Geel, Belgium), Echo Chemical Co., Ltd. (Taichung City, Taiwan) and Merck (Boston, MA, USA). All reactions were performed in oven-dried round-bottom flasks using dry, deoxygenated solvents under argon. A glass–water condenser was fitted over each flask, with a rubber septum fitted over the condenser. Organic solvents were concentrated in rotary evaporators under reduced pressure. All chemical reactions were checked using a Merck TLC (thin-layer chromatography) silica gel 60 F254 plate. Each reaction, including the preparation of starting materials, was purified by flash column chromatography on Merck silica gel 60 (mesh 230–400). Nuclear magnetic resonance spectra (^1^H and ^13^C) were recorded on a Bruker 500 MHz NMR spectrometer. The solvent peaks were used as internal standards and reported as δ values in ppm (chloroform-*d*: 7.25 ppm; methanol-*d_4_*: 3.31, 4.78 ppm; dimethyl sulfoxide-*d_6_*: 2.5 ppm for ^1^H NMR and chloroform-*d*: 77.23 ppm; methanol-*d_4_*: 49.15 ppm; dimethyl sulfoxide-*d_6_*: 39.51 for ^13^C NMR). Multiplicities were recorded as s (singlet), d (doublet), t (triplet), q (quartet), dd (doublets of doublets), m (multiplet) and brs (broad singlet). Data for the ^1^H and ^13^C NMR spectra were reported in terms of the chemical shift relative to TMS (δ = 0.00 ppm). Coupling constants (*J*) were expressed in Hz. HR-MS was performed using a JEOL-HX 110 spectrometer and spectroscopic data were reported as m/z values. Melting points were tested using an electro-thermal instrument.

### 3.1. Chemistry

#### 3.1.1. General Procedure for the Synthesis of Starting Material Ring A Compounds: **A1**–**15**

The corresponding aldehyde (4.0 mmol, 1.0 equiv.), 1- (triphenylphosphoranylidene)-2-propanone (4.8 mmol, 1.2 equiv.) and dry toluene (8 mL) were added to a dry round-bottom flask equipped with a stir bar and condenser. The reaction mixture was stirred and refluxed for two hours. Then, the reaction mixture was concentrated in vacuo. The desired product was purified by chromatography using 10–20% ethyl acetate in hexane to yield the pure corresponding ring A compounds: **A1**–**16** (Appendix B).

#### 3.1.2. General Procedure for the Synthesis of Starting Material Ring B Compounds: **B1**–**4**

Carboxylic acid derivatives (4.0 mmol, 1.0 equiv.) and dry THF (10 mL) were added to a round-bottom flask equipped with a stir bar. The reaction flask was purged with argon and cooled to 0 °C. Then, LiAlH_4_ (4.0 mmol, 1.0 equiv.) was added to the reaction mixture. The reaction was stirred for 1 h. The reaction mixture was quenched with saturated potassium sodium tartrate solution. Upon warming to room temperature, ethyl acetate (10 mL) and celite were added to the reaction mixture to remove unwanted matrices and solids. The solution was filtered with the celite. Then, the filtrate was dried over anhydrous MgSO_4_ and filtered. The filtrate was collected and concentrated in vacuo to give the crude phenyl alcohol compound. The crude phenyl alcohol was dissolved in dichloromethane and pyridinium chlorochromate (4.0 mmol, 1.0 equiv.) was added. The reaction was stirred at room temperature overnight. The resulting mixture was filtered and concentrated in vacuo again. The desired product was purified by chromatography using 10–15% of ethyl acetate in hexane to yield the corresponding ring B compounds: **B1**–**4** (Appendix C).

#### 3.1.3. General Procedure for the Synthesis of β′-Hydroxy-α,β-Unsaturated Ketones (**1**–**8**, **10**–1**7** and **MD1**–**12**) and α,β-Unsaturated Ketones (**1a**, **2a**, **3a**, **4a**, **5a**, **6a**, **7a**, **12a**, **14a** and **MD1a**–**12a**)

The ring A compound (4.0 mmol, 1.0 equiv.) and dry THF (5 mL) were added to a round-bottom flask equipped with a stir bar. The reaction mixture was purged with argon and cooled to −78 °C. Lithium diisopropylamide (4.0 mmol, 1.0 equiv.) was added. The reaction mixture was stirred for 10 min. Then, the ring B compound (4.0 mmol, 1.0 equiv.) in dry THF (5 mL) was added slowly to the ring A-THF solution via syringe. The combined mixture was stirred for 1 h and then quenched with saturated NH_4_Cl solution. The reaction mixture was warmed to room temperature and then extracted with ethyl acetate. The organic phase was separated, dried over anhydrous MgSO_4_ and filtered. The crude material was concentrated in vacuo. The desired product was purified by chromatography using 20–30% ethyl acetate in hexane to produce the corresponding β′-hydroxy-α,β-unsaturated ketones.

The β′-hydroxy-α,β-unsaturated ketone (4.0 mmol, 1.0 equiv.) and dry dichloromethane (10 mL) were added to a round-bottom flask equipped with a stir bar. Then, *p*-toluensulfonic acid (0.4 mmol, 0.1 equiv.) was added to the reaction mixture. The reaction mixture was stirred at 0 °C to room temperature for 30 min. The reaction mixture was quenched with distilled water and then extracted with dichloromethane. The organic phase was separated, dried over anhydrous MgSO_4_ and then filtered. The filtrate was concentrated in vacuo. The desired product was purified by chromatography using 25–30% ethyl acetate in hexane to give the corresponding α, β-unsaturated ketones.

#### 3.1.4. General Procedure for the Synthesis of α,β-Unsaturated β-Diketones **19**–**24** and **MD13**

The β′-hydroxy-α, β-unsaturated ketones **1a**, **2a**, **3a**, **4a**, **6a**, **7a** and **MD4** (4.0 mmol, 1.0 equiv.) in dry dichloromethane (10 mL) were added to a round-bottom flask equipped with a stir bar. Then, the Dess–Martin reagent (4.4 mmol, 1.1 equiv.) was added to the reaction mixture. The reaction mixture was stirred for 30 min and then extracted with dichloromethane. The organic phase was separated, dried over anhydrous MgSO_4_ and then filtered. The filtrate was concentrated in vacuo. The desired product was purified by chromatography using 10–20% ethyl acetate in hexane to give the corresponding α,β-unsaturated β-diketones.

### 3.2. Biology

#### 3.2.1. Cell Cultures

The human cancer cell lines HeLaS3, KBvin, MCF-7, HepG2, NCI-H460 and NCI-H460/MX20 were grown in Dulbecco’s modified Eagle’s medium (DMEM) containing 10% fetal bovine serum (FBS) and 1/100 (*v*/*v*) penicillin/streptomycin at 37 °C under 5% carbon dioxide. The medium was routinely changed every two days. As cell confluence reached 80% in the medium, the cells were treated with 0.25% of trypsin-EDTA for the next passage. Human cervical carcinoma cell line HeLaS3, human non-small-cell lung carcinoma cell line NCI-H460, human liver carcinoma cell line HepG2 and human breast cancer cell line MCF7 were purchased from the Bioresource Collection and Research Center (Hsinchu, Taiwan). The multidrug-resistant human cervical cancer cell line KB-vin was kindly provided by Dr. Kuo-Hsiung Lee (University of North Carolina, Chapel Hill, NC, USA) and maintained with 100 nM of vincristine each week. MDR human non-small-cell lung carcinoma cell line NCI-H460/MX20 was the resistant cell line selected from NCI-H460. It was treated with gradually increasing concentrations of mitoxantrone and maintained in 20 nM of mitoxantrone.

Note: all tested compounds were dissolved in DMSO and the final concentrations of DMSO in all experiments were less than 0.1%.

#### 3.2.2. Anti-Proliferative Assay

Cells (1.5 × 10^5^/mL) were seeded in a 96-well plate overnight, then treated with the synthesized compounds. After the cells were treated with the compounds, MTT solution (1 mg/mL) was added to each well. The plates were incubated for another two hours. Then, the medium was removed. Blue formazan (the yield of reduction in MTT) was dissolved in DMSO. The absorbance was recorded at 570 nm.

#### 3.2.3. Cell Cycle Analysis

Cells (5 × 10^4^ cells/mL) were seeded in a 24-well plate and allowed to adhere overnight. Then, cells were treated with different concentrations of compound **3** and **MD12a** for 24 h and 48 h and fixed with EtOH at 4 °C. After treatment, the cells were collected and washed with PBS. To remove the cells from the medium, trypsin was added. Then, the cells were centrifuged (1500 rpm for 5 min) and washed with PBS. Again, the cells were centrifuged (1500 rpm for 5 min) and the supernatant was removed. After incubation for 15 min, propidium iodine staining buffer (PI: 100 μg/mL) was added. The cell mixture was incubated at 4 °C in the dark for 20 min. The samples were analyzed for propidium iodide DNA fluorescence by flow cytometry (easyCyte 5, Merck Millipore 0500–5005).

#### 3.2.4. Cell Apoptosis Analysis

The cells (5 × 10^4^ cells/well) were seeded in a 24-well plate and allowed to adhere overnight. The seeded cells were then treated with different concentrations of compound **3** and **MD12a** for 24 h and 48 h. After treatment, the cells were collected and washed with PBS, followed by the addition of 1× binding buffer (250 μL) and annexin V-FITC (2.5 μL). The cell mixture was then incubated in the dark at 4 °C for 20 min. Prior to analysis on a flow cytometer, PI (5 μL) was added.

#### 3.2.5. Quantitative PCR Analysis

Cells were prepared by treating with 10 μM of the tested compounds for 48 h. Cells treated with vehicle (DMSO) were used as the controls. A total of 2 μg of total RNA in 14.2 μL nuclease-free H_2_O was used for reverse transcription, using an ABI High-Capacity cDNA Reverse Transcription Kit according to the following, with a total volume of 20 μL: 10 × RT buffer (2 μL), 0.8 μL of dNTP mix (100 mM), 2 μL of RT random primers and 1 μL of MultiScribe™ reverse transcriptase. The reaction was performed with the following program: 25 °C→37 °C (10 min)→85 °C (120 min)→4 °C (5 min) → ∞. The RT products (20 μL) were diluted with 80 μL of nuclease-free H_2_O to generate 5 × dilution RT products (20 ng/μL). Each reaction included 20 ng of cDNA, 500 nM of forward and reverse primers and Bio-Rad iQ™ SYBR^®^ Green Supermix (BioRad, Hercules, CA, USA, 1708880). Total reaction volumes of 10 μL (5.8 μL of SYBR^®^ Green Supermix, 2.2 μL of nuclease-free H_2_O, 1 μL of 5 × dilution RT product (20 ng/μL) and 1 μL of F and R primer (10 μM)) were used. Testing of each sample was repeated three times. A Bio-Rad CFX Connect Real-Time PCR machine was used with the following program: (95 °C→20 s) and 39 cycles of (95 °C→5 s, 60 °C→30 s). The Bio-Rad CFX Manager Version 3.0 software was used for the experimental setup and data analysis. Target gene qPCR data were normalized to the data of a reference gene.

#### 3.2.6. P53 Signaling Pathway Graph

The pathway was drawn using the KEGG PATHWAY Database from Kanehisa Laboratories.

### 3.3. Molecular Modeling

#### 3.3.1. Ligand Preparation

The preparation of the protein was performed using the Prepare Protein module in Discovery Studio 2.5 at the default setting to generate a low-energy 3D conformer, after which the minimized structure of the ligand–compound complex was constructed and finally docked at the drug binding site.

#### 3.3.2. Docking Protocol

The probable drug binding site for the derivative **MD12a** was drawn utilizing the default setting of the Discovery Studio 2.5 software. The protein structure of GADD45B was obtained from the AlphaFold Protein Structure Database [24,25,26,27].

## 4. Conclusions

In our study, 55 new curcuminoid derivatives were synthesized via three-step organic synthesis. The reagents are commercially available. Most derivatives were synthesized in yields of 60% to 90%. The method afforded clean products. The evaluation of the biological activity of the novel derivatives of curcuminoids with the particular scaffolds of α,β-unsaturated β-diketone, β′-hydroxy-α,β-unsaturated ketone and α,β-unsaturated ketone connected with two phenyl groups was performed using an anti-proliferative activity assay and flow cytometry, cell cycle, gene expression and quantitative PCR analyses. Comparing the IC_50_ values of the anti-proliferative activity assay to those of the reference compounds (**1**, **19** and **MD7**) and the 55 new derivatives, **MD12a** and compound **3** were found to be the two most potent compounds, with IC_50_ values of less than 10 μM against all six human cancer cells (HeLaS3, KBvin, MCF-7, HepG2, NCI-H460 and NCI-H460/MX20). The derivatives containing a heterocyclic ring for ring A showed less anti-proliferative activity than the derivatives containing a phenyl group for ring A. Additionally, as the number of methoxy groups on ring A and ring B increased, the anti-proliferative activity of the synthesized compounds such as **MD12a** against these six human cancer cells was increased. In addition, compared to the anti-proliferative activity of α,β-unsaturated β-diketone (enol form), β′-hydroxy-α,β-unsaturated ketone and α, β-unsaturated ketone derivatives, the values of IC_50_ indicated that most derivatives of α,β-unsaturated ketone were more active than the other two forms of the derivatives, except for compound **3**. It is known that α,β-unsaturated ketones contain versatile β-carbon electrophilic sites of carbonyl functional groups at which the nucleophiles can react and form nucleophilic conjugated additions. These derivatives can be deduced and stabilized from resonance structures. Thus, anti-proliferative effects were strongly affected by this specific scaffold of α,β-unsaturated ketone. From the cell cycle analysis, it was found that the application of **MD12a** to HepG2 and MCF-7 cancer cells significantly induced a dose-dependent increase in the sub-G1 phase at 24 h, whereas the application of compound **3** induced a slight time-dependent increase in the sub-G1 phase. In order to observe the phenotypes of cells responding to the tested compounds, images of the changes in cell morphology recorded using a microscope under ×400 magnification showed inflated, deformed, exfoliated and dead cells when increasing concentrations of **MD12a** and compound **3** were applied to HepG2 and MCF-7 cancer cells for 24 h and 48 h. From the flow cytometry data, it can be seen that the application of **MD12a** and compound **3** to HepG2 and MCF-7 cancer cells led to cell apoptosis as the concentration of the tested compounds increased at 24 h and 48 h, especially for **MD12a**. The gene expression array was examined using statistical analysis to obtain histogram plots (showing the normal distribution of the tested groups versus controls), a volcano graph displaying the distribution of up-regulated and down-regulated genes and a heatmap representing the distribution of up-regulated and down-regulated genes determined by clustering analysis from the gene expression profile for the different samples and treatment conditions. Using a gene expression array, the possible pathways and related genes were discovered. The quantitative PCR analysis displayed the fold changes of three related genes (GADD45B, SENS2 and BBC3) for **MD12a** and compound **3**, which were applied to HepG2 and MCF-7 cancer cells. It was discovered that **MD12a** highly up-regulated the expression of GADD45B for MCF-7 cells in the p53 pathway, according to the fold change analysis shown in Figure 10 based on the gene expression array (genes shown in red indicate a strong response in the pathway). Therefore, **MD12a** significantly up-regulated the expression of GADD45B, leading to the suppression of cell formation, apoptosis and cell death. From the docking analysis, the data also showed that the interactions between GADD45B and **MD12a**, including one hydrogen bond and three hydrophobic interactions, could stabilize the ligand–compound complex at the optimized conformation of the drug binding site. Compared to the other novel derivatives, **MD12a** with a promising scaffold of α,β–unsaturated ketone was found to be the most potent curcuminoid derivative against all six human cancer cells in our research. The evaluation of its cytotoxicity showed that **MD12a** is a promising compound that should be used for the development of targeted treatments of cancer cells in further research.

## Data Availability

Not applicable.

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
