# Peer review of "Design, Synthesis and Evaluation of Novel Derivatives of Curcuminoids with Cytotoxicity"

_ijms, 2021, doi:10.3390/ijms222212171_

Round 1
Reviewer 1 Report
Curcuminoids are the promising natural compound with a large variety of therapeutic properties. Unfortunately the clinical applications of curcuminoids are restricted by their poor solubility, low absorption and bioavailability, high metabolism rate. The authors present in their work the synthesis and anticancer activity of new curcumin derivatives. The work is divided into two parts: chemical and biological. However, the first part on synthesis is incomplete. There are no basic data characterizing the obtained new compounds, such as NMR, MS, HRMS, mp, IR and yields. In para. 4 the authors only admit that such measurements were made, and for unknown reasons they explain the meaning of abbreviations that they do not use in their work anyway (s = singlet, d = doublet ...). Such a significant shortage disqualifies the works for publication. Other remarks that make the work difficult to follow include, inter alia, the wrong and complicated numbering of compounds, tables. Here are some examples:
- In Schemes 2 and 3, two different compounds have the same numbers 1a-7a, MD24. The introduction of MD numbering is incomprehensible and confusing.
- In schemes, Ring B should be replaced by Ph since Ring B = Ph.
- Table 1 (p. 3) shows that since R2 = Ph, there is a Ph-Ph bond in the obtained compounds. Should the Ph in Table 1 (p.3) rather be replaced by H ?
- Where is the Table A (line 58)? Why are the table on page 3 and the table on page 6 the same number?
- In Scheme 1, the final products have no numbers at all.
- On what basis was only the enol form of the compound in the Scheme 3 assumed?
- The authors write about less-cost and less time-consuming synthesis but do not provide examples of alternative methods for comparison. (line 448)
Examples of incorrect phrases:. "... was added to the Part A solution."; "The compound 18 was a minor side product of 6a." "... organic synthesis with high productivity for most curcuminoid derivatives is focused on our research."
Despite the undoubtedly large amount of work put into the preparation of the manuscript, it still requires extensive revision to be accepted for publication.
Author Response
Dear reviewer,
We would like to thank you and the reviewers for the careful and thorough reading of this manuscript and for the invaluable comments. The corresponding comments were refined in the revised paper and our responses were summarized below. Also, please see the attachment for further information.
#1 Comments and Suggestions for Authors
Curcuminoids are the promising natural compound with a large variety of therapeutic properties. Unfortunately, the clinical applications of curcuminoids are restricted by their poor solubility, low absorption and bioavailability, high metabolism rate. The authors present in their work the synthesis and anticancer activity of new curcumin derivatives. The work is divided into two parts: chemical and biological. However, the first part on synthesis is incomplete. There are no basic data characterizing the obtained new compounds, such as NMR, MS, HRMS, mp, IR and yields. In para. 4 the authors only admit that such measurements were made, and for unknown reasons they explain the meaning of abbreviations that they do not use in their work anyway (s = singlet, d = doublet ...). Such a significant shortage disqualifies the works for publication. Other remarks that make the work difficult to follow include, inter alia, the wrong and complicated numbering of compounds, tables. Here are some examples:
-- In Schemes 2 and 3, two different compounds have the same numbers 1a-7a, MD24. The introduction of MD numbering is incomprehensible and confusing.
Response:
Thank you so much for your comments. We made the change for the above errors you mentioned. For the NMR Spectra and HRMS, please see the supplementary materials.
Because we discuss these synthesized compounds in three different aspects according to the anti-proliferative effects:
- compare aromatic rings: heterocyclic ring vs. phenyl
- compare the scaffold of compounds that beta’-hydroxy-alpha, beta unsaturated-ketone compounds changed to alpha, beta-unsaturated ketone and alpha, beta-unsaturated diketone compounds.
- Finally, modify the both rings with different number of -OMe group. That’s why we categorized these compounds in MD series.
In addition, in order to make more clear for readers, the compounds also gave the abbreviation according to their skeletons, so that it would be easier to distinguish these compounds.
-- In schemes, Ring B should be replaced by Ph since Ring B = Ph.
- Table 1 (p. 3) shows that since R2 = Ph, there is a Ph-Ph bond in the obtained compounds. Should the Ph in Table 1 (p.3) rather be replaced by H ?
Response:
Thank you for the comments. We revised the scheme and table. Please see the manuscript.
-- Where is the Table A (line 58)? Why are the table on page 3 and the table on page 6 the same number?
Response:
Thank you for the comments. We made the change for the number of table. Please see the manuscript.
-- In Scheme 1, the final products have no numbers at all.
Response:
Thank you. We drew the new scheme. Please see the manuscript.
-- On what basis was only the enol form of the compound in the Scheme 3 assumed?
Response:
Based on 13C NMR spectra, there is only one peak at around 201 ppm indicating one carbonyl group, whereas a peak at around 177-179 ppm indicated –OH on beta’ carbon position. According to 1H NMR spectra, the chemical shift of –H on alpha’ carbon position was measured at around 5.6 ppm (Please refer to the NMR spectra on supplementary materials).
In addition, the keto form and enol form are interconvertible. Enol form is more stable due to the presence of continuous conjugated system and intramolecular hydrogen-bonding according to the reference 12.

Reviewer 2 Report
The manuscript “Design, synthesis and evaluation of novel derivatives of curcuminoids for anticancer activity” by Chen et al. dials with a high productivity and speedy synthesis of curcuminoid derivatives and in vitro evaluation of their anticancer activity.
It seems that most of the authors are chemists, because the chemical part of the manuscript is fine in contrast to the biological evaluation, where significant improvements are required.
From methodological point of view:
1. It is not clear in what solution(s) were dissolved the tested compounds, and this solution(s) has to be used in equal concentration(s) as a negative control instead non-treated cells.
2. To evaluate “anticancer activity” authors have to include non-cancer, normal cell lines as a control, to be able to compare the results with this control. Otherwise, they can evaluate cytotoxicity, but not anticancer activity. In this manuscript, the authors did not use such control.
3. The authors have to explain why they use these cell lines? For the specialists is clear that some of the cell lines are regular cancer cells, and others are used as drug-resistant models, but for most readers this is not known. Even more, the origin and the type of the cells is not given in the section “Materials and methods” (4.2.1).
4. The authors have to provide in 4.2.2. in what concentration the cells were seeded.
5. In section 4.2.3. Cell cycle analysis, the used concentration of the cells for this analysis (5x104 cells/well) in 24-well plate, is too low. PI binds to DNA, but also binds to RNA, and the cells have to be treated with RNase to degrade RNA before the PI staining. This was not conducted. It is written “Then cells were treated with different concentrations of compound 3 and MD24 577 and fixed with EtOH at 4°C for 24 h and 48 h.”, which is not correct. The cells should be treated for these periods, but not fixed for 24 and 48 h.
6. It is not clear when is performed the Quantitative PCR analysis – before the treatment, after treatment for 24 h, after treatment for 48 h, and what concentration is used for this treatment?
7. How was analyzed the cell morphology?
8. It is not clear why the authors performed molecular modeling.
9. There are many papers related to the curcumin and curcuminoid derivatives, and the respective activities. In this manuscript, there is not any discussion or comparison for the known and the new curcuminoid derivatives
10. The English has to be improved, especially in the abstract, lines 87-100
Results:
The presentation of the results is duplicated – in tables and figures. The authors have to choose only one form of presentation.
For example, Figures 2a1 and 2a2 duplicate the data from Table 2a; Figures 2b1 and 2b2 duplicate the data from Table 2b; Figures 2c1 and 2c2 duplicate the data from Table 2c; Figures 2d1 and 2d2 duplicate the data from Table 2d.
2.2.3 Cell Morphology – It is described very briefly. What means “unshaped cells”? (line 317). Each cell has a shape. How were detected the dead cells? There is not a scale bar on the photos. Only the magnification in which are taken the pictures is given.
Histogram Plot and Vulcano plot are not explained. The text is like a legend. What conclusions suggest these figures?
Figure 10 (P53 signaling pathway) is complicated and does not provide an important information about the role of GADD45B, which is essential only for MCF-7 cells treated with MD24, where the control (see point 1) may be is not accurate.
Author Response
Dear Editor,
We would like to thank you and the reviewers for the careful and thorough reading of this manuscript and for the invaluable comments. The corresponding comments were refined in the revised paper and our responses were summarized below.
#2 Comments and Suggestions for Authors
The manuscript “Design, synthesis and evaluation of novel derivatives of curcuminoids for anticancer activity” by Chen et al. dials with a high productivity and speedy synthesis of curcuminoid derivatives and in vitro evaluation of their anticancer activity.
It seems that most of the authors are chemists, because the chemical part of the manuscript is fine in contrast to the biological evaluation, where significant improvements are required.
From methodological point of view:
- It is not clear in what solution(s) were dissolved the tested compounds, and this solution(s) has to be used in equal concentration(s) as a negative controlinstead non-treated cells.
Response:
Thank you for the comment. The compounds were dissolved in DMSO and it was used as vehicle control in non-treated cells.
- To evaluate “anticancer activity” authors have to include non-cancer, normal cell lines as a control, to be able to compare the results with this control. Otherwise, they can evaluate cytotoxicity, but not anticancer activity. In this manuscript, the authors did not use such control.
Response:
Thank you for the suggestion. We have corrected “anticancer activity” to “cytotoxicity” in the revised manuscript according to the comment.
- The authors have to explain why they use these cell lines? For the specialists is clear that some of the cell lines are regular cancer cells, and others are used as drug-resistant models, but for most readers this is not known. Even more, the origin and the type of the cells is not given in the section “Materials and methods” (4.2.1).
Response:
Thank you for the comment. We have provided details of the cell lines used in the present study “Materials and methods” (4.2.1).
Materials and methods (4.2.1):
Human cervical carcinoma cell line HeLaS3, human non-small cell lung carcinoma cell line NCI-H460, human liver carcinoma cell line HepG2 and human breast cancer cell line MCF7 were purchased from Bioresource Collection and Research Center (Hsinchu, Taiwan). The multidrug resistant human cervical cancer cell line KB-vin was kindly provided by Dr. Kuo-Hsiung Lee (University of North Carolina, Chapel Hill, U.S.A) and maintained with 100nM vincristine per week. MDR human non-small cell lung carcinoma cell line NCI-H460/MX20 was the resistant cell line selected from NCI-H460 with gradually increased concentrations of mitoxantrone and maintained in 20 nM mitoxantrone.
- The authors have to provide in 4.2.2. in what concentration the cells were seeded.
Response:
Thank you and we have added the concentration of the cells in 4.2.2. “1.5x105/mL cells were seeded in 96-well plate……”
- In section 4.2.3. Cell cycle analysis, the used concentration of the cells for this analysis (5x104cells/well) in 24-well plate, is too low. PI binds to DNA, but also binds to RNA, and the cells have to be treated with RNase to degrade RNA before the PI staining. This was not conducted. It is written “Then cells were treated with different concentrations of compound 3 and MD24 577 and fixed with EtOH at 4°C for 24 h and 48 h.”, which is not correct. The cells should be treated for these periods, but not fixed for 24 and 48 h.
Response:
Thank you for the comment. The cell concentration was 5x104 cells/mL not 5x104 cells/well. We have corrected in the revised manuscript. In addition, the PI solution we used in the present study contained RNase, therefore, we did not add extra RNase during the experiment.
We have also corrected the sentence “Then cells were treated with different concentrations of compound 3 and MD24 and fixed with EtOH at 4°C for 24 h and 48 h.” to “ Then cells were treated with different concentrations of compound 3 and MD24 for 24 h and 48 h.and fixed with EtOH at 4°C.”
- It is not clear when is performed the Quantitative PCR analysis – before the treatment, after treatment for 24 h, after treatment for 48 h, and what concentration is used for this treatment?
Response:
Please refer to the revision of manuscript. Thank you for the comments.
- How was analyzed the cell morphology?
Response:
Please refer to the revision of manuscript. Thank you for the comments
- It is not clear why the authors performed molecular modeling.
Response:
The purpose we performed molecular modeling is that we like to know the interaction of between the tested compound MD24.
- There are many papers related to the curcumin and curcuminoid derivatives, and the respective activities. In this manuscript, there is not any discussion or comparison for the known and the new curcuminoid derivatives
Response:
Design and develop less steps to synthesize compounds is the first major work. Several methods we have tried to prepare for starting material costed a lot of time and wasted a lot of solvents for purification. Some preparations for starting materials ran into problems because it was difficult to separate the major product from side products. Finally, the low yield of product was obtained. It was not practical and also polluted the environment when solvents were evaporated in the air if many steps were taken. Compared to the reference 12, the yield of most products was low. The price of the starting materials is very expensive.
- The English has to be improved, especially in the abstract, lines 87-100
Response:
Thank you very much for your suggestions. We have revised the abstract and English revision and manuscript layout will be provided by MDPI if the article is accepted.
Results:
The presentation of the results is duplicated – in tables and figures. The authors have to choose only one form of presentation.
- For example, Figures 2a1 and 2a2 duplicate the data from Table 2a; Figures 2b1 and 2b2 duplicate the data from Table 2b; Figures 2c1 and 2c2 duplicate the data from Table 2c; Figures 2d1 and 2d2 duplicate the data from Table 2d.
Response:
We have replaced the figure to table. Thank you.
- 2.3 Cell Morphology – It is described very briefly. What means “unshaped cells”? (line 317). Each cell has a shape. How were detected the dead cells? There is not a scale bar on the photos. Only the magnification in which are taken the pictures is given.
Response:
At that time, we’d just simply like to know the phenotype change for cells by treating with new compounds and recorded the difference between tested compounds.
- Histogram Plot and Volcano plot are not explained. The text is like a legend. What conclusions suggest these figures?
Response:
Thank you for the comment. Please refer to the conclusion part of revised manuscript. “….. The gene expression array was examined by the statistical analysis to obtain histogram plots (the normal distribution of tested groups versus controls), the volcano graph dis-playing the distribution of up-regulated and down-regulated genes, and heatmap repre-senting distribution of up- and down-regulated genes examined by clustering analysis from gene expression profile for samples and treatment conditions. By gene expression array, the possible pathways and related genes were discovered….”
- Figure 10 (P53 signaling pathway) is complicated and does not provide an important information about the role of GADD45B, which is essential only for MCF-7 cells treated with MD24, where the control (see point 1) may be is not accurate.
Response:
Thank you for the comment. Please refer to the conclusion on the revised manuscript. The figure10 has been changed to figure 7 (the new figure which is much clear to answer the question).

Round 2
Reviewer 1 Report
The manuscript has serious flaws and still requires a thorough redrafting, because in its current form the reader is doomed to be led astray. Descriptions of spectra in SM should be carefully checked as they are incorrect. In some descriptions of 1H NMR spectra even 10H is missing! Particular attention should be paid to the style and spelling of the language used, and the numbering of the compounds. The revised manuscript, despite earlier corrections, cannot be published in my opinion. I would like to support this opinion with a few more examples, therefore some errors and inaccuracies are presented below.
- The compound numbers should not be mentioned in the abstract.
- The form of the caption for the fig. 1 is not correct. The descriptive part should be included in the main text.
- line 62. The cpd 18 is missing in the scheme 1.
- The abbr. MD (methoxy derivatives?) requires explanation.
- Maybe it will be more logical and clear to replace the MD1, MD2, MD3, MD4 ... series numbers with MD1, MD1a, MD2, MD2a ... according to the numbering 1,1a, 2, 2a ...
- The MD series requires a clear marking for each number: MD1, MD2, MD5 etc., not MD1, 2, 5
- Scheme 2 can be removed because it is a duplication of a part of scheme 1.
- In the scheme 1 “alpha”, “beta” should be replaced with “α”, “β”. Also the numbers 1a-14a can be replaced with 1a-7a, 12a, 14a.
- Compounds 9 and 18 in the Table 1 and compounds 9,18 in the SM are different compounds!
- The experimental part needs to be improved. e.g. correct the style, the description of the spectra (SM), the procedures do not contain the precise number of moles.
- The compounds and procedures already described in the literature should be referenced.
- line 523 “Ethyl acetate (10 ml) and Celites were added into the mixture. Celites were filtered out.” The use of Celite in this case is unclear.
- Examples of incorrect phrases:
„ring A (B) compounds” „measured on Table 2“
line 14-15.„ The research and development of production for 14 curcuminoids by less steps of organic synthesis is focused on our work”
line53 Carboxaldehyde should be replaced with aldehyde.
line 51 „novel compounds of β’-hydroxy-α,β-unsaturated ketone” change to “novel β’-hydroxy-α,β-unsaturated ketones”. There should also be: ketones, diketones, etc.
line 86. Should be as follows: “: …hydroxy group was detected…
The phrase „ketone compounds” should be replaced with “carbonyl compounds” or „ketones”
etc.
Author Response
Dear Reviewer,
We would like to thank you for the careful and thorough reading of this manuscript and for the invaluable comments. The corresponding comments were refined in the revised paper and our responses were summarized below. Also, please refer to the revision manuscript and supplementary materials.
Comments and suggestions for reviewer#1
The manuscript has serious flaws and still requires a thorough redrafting, because in its current form the reader is doomed to be led astray. Descriptions of spectra in SM should be carefully checked as they are incorrect. In some descriptions of 1H NMR spectra even 10H is missing! Particular attention should be paid to the style and spelling of the language used, and the numbering of the compounds. The revised manuscript, despite earlier corrections, cannot be published in my opinion. I would like to support this opinion with a few more examples, therefore some errors and inaccuracies are presented below.
- The compound numbers should not be mentioned in the abstract.
Response:
Thank you for the suggestions. The abstract has been revised. We put the chemical names for the two potent compounds. We have considered that the chemical name for curcuminoid is too long so that the reader may be confused. Therefore, we still decided to put the compound numbers after their chemical names. Please refer to the manuscript.
- The form of the caption for the fig. 1 is not correct. The descriptive part should be included in the main text.
Response:
Thank you for the comments. The descriptions of fig. 1 have been added to main text. Please refer to the manuscript.
- line 62. The cpd 18 is missing in the scheme 1.
Response:
Thank you for the comments. The compound 18 (5.3% yield) is a side product of compound 6. It might be happened when compounds were purified through the silica gel column. The beta’-hydoxy was replaced to beta’-chloride. In order to solve the problem, we need to make sure to use the neutral silica gel to do the purification for compounds. Therefore, the compound 18 was excluded from the scheme 1. The reason also mentioned in the main text. Also, the reason we include compound 18 in the table 1 and 2 because we have recorded its cell anti-proliferative effects.
- The abbr. MD (methoxy derivatives?) requires explanation.
Response:
Thank you for the suggestions. Indeed, the abbreviation of MD means the different numbers of methoxy groups on ring A and ring B of curcumoind derivatives. These modified compounds were categorized in MD series. The explanation was added into the manuscript.
- Maybe it will be more logical and clear to replace the MD1, MD2, MD3, MD4 ... series numbers with MD1, MD1a, MD2, MD2a ... according to the numbering 1,1a, 2, 2a ...
Response:
Thank you for the suggestions. The labels of compounds have been changed to MD1, MD1a, MD2, MD2a… Please refer to the manuscript.
- The MD series requires a clear marking for each number: MD1, MD2, MD5 etc., not MD1, 2, 5
Response:
Thank you for your suggestions. We revised and indicated each number for compounds clearly. Please refer to the revision manuscript.
- Scheme 2 can be removed because it is a duplication of a part of scheme 1.
Response:
The scheme 2 have been removed. Please refer to the revision manuscript. Thank you so much for the suggestions.
- In the scheme 1 “alpha”, “beta” should be replaced with “α”, “β”. Also the numbers 1a-14a can be replaced with 1a-7a, 12a, 14a.
Response:
The “alpha”, “beta” on the scheme 1 have been changed to “α”, “β”. Please refer to the revision manuscript. Thank you so much for the suggestions.
- Compounds 9 and 18 in the Table 1 and compounds 9,18 in the SM are different compounds!
Response:
Thank you for the comments. For more explanation, please refer to the manuscript. Compound 9 was also excluded from the scheme, but we have recorded its anti-proliferative data. Therefore, compound 9 were included in the table 1 and 2. The data of NMR and HRMS also confirmed the structure of compound 9 and 18.
- The experimental part needs to be improved. e.g. correct the style, the description of the spectra (SM), the procedures do not contain the precise number of moles.
Response:
Thank you for the suggestions. We corrected the style, description of spectra, and put the precise number of moles in the procedures. Please refer to the manuscript.
Also, about the spectra description, if we want to assign which proton is which precisely in the supplementary materials unless we did more NMR analysis for the compounds, such as COSY, HSQC, HMBC… So, we would like to simply put the peak, coupling constant, number of protons in the NMR description.
- The compounds and procedures already described in the literature should be referenced.
Response:
The procedure in the literature (the section of 2.1 chemistry) and reference compounds (in the table 1 and supplementary materials) have been referenced. Please refer to the manuscript.
- line 523 “Ethyl acetate (10 ml) and Celites were added into the mixture. Celites were filtered out.” The use of Celite in this case is unclear.
Response:
Thank you for your comments. The use of celites in this experiment was to remove unwanted matrices and solids. We added the explanation into the section of materials and methods.
- Examples of incorrect phrases:
„ring A (B) compounds” „measured on Table 2“
line 14-15.„ The research and development of production for 14 curcuminoids by less steps of organic synthesis is focused on our work”
line53 Carboxaldehyde should be replaced with aldehyde.
line 51 „novel compounds of β’-hydroxy-α,β-unsaturated ketone” change to “novel β’-hydroxy-α,β-unsaturated ketones”. There should also be: ketones, diketones, etc.
line 86. Should be as follows: “: …hydroxy group was detected…
The phrase „ketone compounds” should be replaced with “carbonyl compounds” or „ketones”
etc.
Response:
Thank you for the suggestions. The incorrect phrases descripted above have been revised. Also, the manuscript has been send to English revision (English editing ID: english-35539). Please refer to the revision manuscript.

Reviewer 2 Report
The authors significantly improved the manuscript, but for some of the notes they gave answers without changing or including the relevant information in the manuscript. For example, they answered to the first note that the compounds are dissolved in DMSO, but this is not included in the manuscript. In the section “Materials and Methods” it should be written that “All tested compounds were dissolved in DMSO”. Also, in this section there is not information about the used concentrations and volume of the tested compounds for the treatment. It is important for the readers to know the concentration and the volume of the added test-compounds as well as the DMSO control. It is important what was the final concentration of the DMSO.
Some information was added in the section Results instead in section “Materials and Methods” (lines 290-294 related to the cell morphology and lines 408-417 related to the molecular modeling).
Author Response
Dear Reviewer,
We would like to thank you for the careful and thorough reading of this manuscript and for the invaluable comments. The corresponding comments were refined in the revised paper and our responses were summarized below.
Comments and suggestions for reviewer#2
The authors significantly improved the manuscript, but for some of the notes they gave answers without changing or including the relevant information in the manuscript. For example, they answered to the first note that the compounds are dissolved in DMSO, but this is not included in the manuscript. In the section “Materials and Methods” it should be written that “All tested compounds were dissolved in DMSO”. Also, in this section there is not information about the used concentrations and volume of the tested compounds for the treatment. It is important for the readers to know the concentration and the volume of the added test-compounds as well as the DMSO control. It is important what was the final concentration of the DMSO.
Some information was added in the section Results instead in section “Materials and Methods” (lines 290-294 related to the cell morphology and lines 408-417 related to the molecular modeling).
Response:
Thank you very much for your suggestions. It’s really helpful for us for either this work or the further research. We revised and added the information (all tested compounds were dissolved in DMSO and the final concentrations of DMSO in all experiments were less than 0.1%) in the main text and the section of materials and methods. We also send our manuscript for English editing (English editing ID: english-35539). Please refer to the revision manuscript.

Round 3
Reviewer 1 Report
Third review.
The manuscript still requires thorough adjustments. Some of the errors have already been indicated, unfortunately the authors have not corrected them.
- Line 42
The correct form is: “These active compounds derived from turmeric are named curcuminoids, a kind of diarylheptanoids which have a linear di-phenol structure and α,β-unsaturated β-diketone moiety”
- Line 84
“Compound 18 (a chloride at the C5 beta position) was a side product of compound 6. The result of replacing the hydroxy at C5 with chloride might occur when the compounds are purified by silica gel column chromatography. In order to solve this problem, a neutral silica gel column was obtained and utilized during purification.” – Did the authors use a "chlorinating gel"? Where does chlorine come from? The authors should explain it differently because the explanation given is improbable.
- The main text should include the structures of compounds 9 and 18.
- The compounds 9 and 18 should be removed from table 1. (Change the title of the table).
- Line 114
Should be as follows: „… different numbers of methoxy or hydroxy groups on each phenyl rings A and B are categorized as MD series.”
- Replace “hydroxy” with “hydroxy group”.
- Ref. [12] should be changed as it does not apply to the MD7 compound (Table 1).
- Line 94 and 488. “Produced” should be changed to “synthesized”.
- Supplementary Materials:
Wrong amount of hydrogen atoms in the description of 1H NMR spectra (e.g. for comp. 2, as many as 10 H are missing).
HRMS does not correspond to the proposed compound structure (e.g. MD5).
Author Response
Dear Reviewer,
We would like to thank you for the careful and thorough reading of this manuscript and for the invaluable comments. The corresponding comments were refined in the revised paper and our responses were summarized below.
Comments and suggestions for reviewer#1
Third review.
The manuscript still requires thorough adjustments. Some of the errors have already been indicated, unfortunately the authors have not corrected them.
- Line 42
The correct form is: “These active compounds derived from turmeric are named curcuminoids, a kind of diarylheptanoids which have a linear di-phenol structure and α,β-unsaturated β-diketone moiety”
Response:
Thank you for the suggestion. We corrected the description. Please refer to the manuscript.
- Line 84
“Compound 18 (a chloride at the C5 beta position) was a side product of compound 6. The result of replacing the hydroxy at C5 with chloride might occur when the compounds are purified by silica gel column chromatography. In order to solve this problem, a neutral silica gel column was obtained and utilized during purification.” – Did the authors use a "chlorinating gel"? Where does chlorine come from? The authors should explain it differently because the explanation given is improbable.
Response:
Thank you for the suggestion. We corrected the formation of compound 18. “Compound 18 (a chlorine substituent at the C5 beta position), a side product of compound 6, might be obtained during the acid workup with ammonium chloride.”
- The main text should include the structures of compounds 9 and 18.
Response:
The structures of compound 9 and 18 labeled as the side products were added into the scheme 1.
- The compounds 9 and 18 should be removed from table 1. (Change the title of the table).
Response:
The compound 9 and 18 were removed from the table 1 and the title were changed.
- Line 114
Should be as follows: „… different numbers of methoxy or hydroxy groups on each phenyl rings A and B are categorized as MD series.”
Response:
Thank you for your suggestion. We corrected the description. Please refer to the manuscript.
- Replace “hydroxy” with “hydroxy group”.
Response:
Thank you for your comments. We corrected it.
- [12] should be changed as it does not apply to the MD7 compound (Table 1).
Response:
Thank you very much for your suggestion. We changed the reference 12 to “Chemical and Pharmaceutical Bulletin, 2011, vol. 59, # 8, p. 1042 – 1044” indicating that (E)-1,7-bis(3,4-dimethoxyphenyl)-5-hydroxyhept-1-en-3-one (MD7) was first synthesized by multi-step reaction:
- Line 94 and 488. “Produced” should be changed to “synthesized”.
Response:
Thank you for the comments. We revised it.
- Supplementary Materials:
Wrong amount of hydrogen atoms in the description of 1H NMR spectra (e.g. for comp. 2, as many as 10 H are missing).
HRMS does not correspond to the proposed compound structure (e.g. MD5).
Response:
Thank you very much for the suggestion. We correct the errors. The amount of H atoms for compound 2 was recalculated by 1HNMR data. For the compound MD5, we revised the mass data.
